# Regulation of innate immune responses in macrophages differentiated in the presence of vitamin D and infected with dengue virus 2

Jorge Andrés Castillo[1,2], Diana M. Giraldo[1], Juan C. Hernandez[3], Jolanda M. Smit[2], Izabela A. Rodenhuis-Zybert[2], Silvio Urcuqui-Inchima[1]*

1 Grupo de Inmunovirología, Departamento de Microbiología y Parasitología, Facultad de Medicina, Universidad de Antioquia UdeA, Medellín, (Antioquia), Colombia, 2 Department of Medical Microbiology and infection Prevention, University of Groningen and University Medical Center Groningen, Groningen, The Netherlands, 3 Infettare, Facultad de Medicina, Universidad Cooperativa de Colombia, Medellín, (Antioquia), Colombia

* silvio.urcuqui@udea.edu.co

**Data Availability Statement:** All relevant data are within the manuscript and its Supporting Information files.

## Abstract

A dysregulated or exacerbated inflammatory response is thought to be the key driver of the pathogenesis of severe disease caused by the mosquito-borne dengue virus (DENV). Compounds that restrict virus replication and modulate the inflammatory response could thus serve as promising therapeutics mitigating the disease pathogenesis. We and others have previously shown that macrophages, which are important cellular targets for DENV replication, differentiated in the presence of bioactive vitamin D (VitD3) are less permissive to viral replication, and produce lower levels of pro-inflammatory cytokines. Therefore, we here evaluated the extent and kinetics of innate immune responses of DENV-2 infected monocytes differentiated into macrophages in the presence ($D_3$-MDMs) or absence of VitD3 (MDMs). We found that $D_3$-MDMs expressed lower levels of RIG I, Toll-like receptor (TLR) 3, and TLR7, as well as higher levels of SOCS-1 in response to DENV-2 infection. $D_3$-MDMs produced lower levels of reactive oxygen species, related to a lower expression of TLR9. Moreover, although VitD3 treatment did not modulate either the expression of IFN-α or IFN-β, higher expression of protein kinase R (PKR) and $2'$-$5'$-oligoadenylate synthetase 1 (OAS1) mRNA were found in $D_3$-MDMs. Importantly, the observed effects were independent of reduced infection, highlighting the intrinsic differences between $D_3$-MDMs and MDMs. Taken together, our results suggest that differentiation of MDMs in the presence of VitD3 modulates innate immunity in responses to DENV-2 infection.

## Author summary

Dengue virus (DENV) accounts for one of the most transmitted mosquito-borne infectious diseases worldwide. While an estimated 390 million infections occur per year, no specific antiviral compounds are available yet. Further, Denvaxia approved vaccine is not fully protective. Research of new therapies that could control infection caused by DENV is

**Funding:** This work was funded by the Minciencias/Colciencias [grant No. 111584467188 contract No. 595 de 2019] to SUI and Universidad de Antioquia-CODI Acta No. 2015-7803 to SUI. The funders had no role in study design, data collection and analysis, decision to publish, or preparation of the manuscript.

**Competing interests:** The authors have declared that no competing interests exist.

thus important to the field. Also, therapies that can modulate the inflammatory immune response, critical for severe progression in DENV-infected patients, may represent good candidates for fighting DENV infections. Here, we continued the research that we have been doing the last years, regarding the antiviral and immunomodulatory properties of Vitamin D (VitD3) against DENV infection in human macrophages, which represent a natural target for DENV replication. We show that VitD3 modulates innate immune responses of macrophages in response to DENV 2 infection, through i) downregulation of TLRs ii) decrease in ROS production, and iii) upregulation of SOCS 1 and IFN-stimulated genes such as PKR and OAS. Our study increases our insights into the mechanisms underlying the antiviral and anti-inflammatory effects of VitD3 in DENV-2 infected macrophages, which ultimately would contribute to the progress of VitD3 as a candidate for anti-DENV therapy.

## Introduction

With an estimated 390 million infections every year, the dengue virus (DENV) is one of the most frequently transmitted arboviruses in tropical and subtropical countries [1]. Although close to 75% of DENV infection cases are asymptomatic, the clinically apparent infections can range from a mild flu-like condition (dengue) to a life-threatening (severe dengue) disease characterized by hemorrhages and hypovolemic shock [2]. To date, predicting the disease outcome is impossible and there is no specific treatment available. In addition, Denvaxia, the only available vaccine approved in 11 countries, is only partially effective [3]. The availability of an effective therapeutic strategy is, therefore, an important goal, considering that almost 50% of the human population lives in zones with a risk of transmission [1]. The mechanism underlying the aberrant inflammatory response is not fully understood, however, the infection-induced excessive production of proinflammatory cytokines is known to promote endothelial permeability and damage to endothelial integrity, thought to be responsible for severe disease [4]. Consequently, ameliorating exacerbated inflammation is considered a key aim of therapeutic strategies.

Immune sentinels such as monocytes, dendritic cells (DCs), and macrophages sense DENV through an array of pattern recognition receptors (PRRs) including retinoic acid-inducible gene-I-like receptors (RLRs) such as RIG-I and MDA5 [5], and Toll-like receptors (TLRs) [6,7]. Recognition of molecular patterns expressed in DENV virus or replication intermediates by PRRs ultimately leads to the production of proinflammatory cytokines and antiviral type I interferons (IFN I), which play a crucial role in containing the infection. Importantly, DENV can also replicate in immune cells and has evolved mechanisms to alter or evade innate immune responses (reviewed in [8]. Indeed, aberrant inflammation characterized by inflammatory cytokines predominating over the antiviral response is thought to lead to endothelial permeability underlying severe disease [9,10].

Given the importance of the inflammatory response in the pathogenesis of DENV infection, the use of immune-regulatory compounds such as vitamin D (VitD3) could be useful as an alternative therapeutic strategy for DENV-infected patients. In fact, this hormone shows several immunomodulatory effects in various cells of the immune system, including antigen-presenting cells [11]. Furthermore, VitD3 has an antiviral activity that impairs the replication of several viruses including the hepatitis C virus (HCV), the human immunodeficiency virus (HIV), and the respiratory syncytial virus (RSV) and DENV [12–14]. In vitro studies have shown that VitD3 decreases DENV infection and replication and the production of proinflammatory cytokines [15,16]. We have previously demonstrated that monocyte-derived macrophages (MDMs) differentiated in the presence of VitD3 ($D_3$-MDMs) are more resistant to

DENV-2 infection, due to a decreased expression of mannose receptor (MR) and thus reduced viral uptake [17]. Likewise, the production of proinflammatory cytokines in response to DENV infection was decreased in $D_3$-MDMs [17]. In addition, we reported a decrease in the susceptibility to DENV-2 infection and the production of proinflammatory cytokines by MDMs obtained from healthy individuals who received a high-dose oral supplementation of VitD3 for 10 days [18]. Indeed, VitD3 has been shown to downregulate the expression of several TLRs in monocytes and thus contribute to the amelioration of the inflammatory response triggered by IFN-γ, LPS, or lipoteichoic acid [19,20]. Also, it has been shown that VitD3 suppressed the inflammatory response of LPS in mice by up-regulating the expression of the suppressor of cytokine signaling 1 (SOCS-1) [21], an important regulator of inflammatory responses [22]. This body of evidence demonstrates that VitD3 has wide immunomodulatory and antiviral activities against DENV infection.

Consequently, the present study aimed to evaluate the effect of VitD3 on the regulation of the innate immune responses during DENV infection. To do so, we assessed temporal changes in the expression of the most studied PRRs, SOCS-1, IFN I, and IFN-stimulated genes (ISGs) in MDMs and $D_3$-MDMs infected with DENV-2. Altogether, the results of this study increase our insights into the mechanisms underlying the antiviral and anti-inflammatory effects of VitD3 in DENV-2 infected macrophages.

## Materials and methods

### Ethics statement

The protocols for individual enrollment and sample collection were approved by the Committee of Bioethics Research of Sede de Investigación Universitaria, Universidad de Antioquia (Medellín, Colombia), and inclusion was preceded by a signed informed consent form, according to the principles expressed in the Declaration of Helsinki.

### Cells and reagents

The mosquito C6/36 HT cell line was obtained from the American Type Culture Collection (ATCC) and cultured in Leibovitz L-15 medium (Sigma Aldrich, USA) supplemented with 10% v/v heat-inactivated fetal bovine serum (FBS) (Thermo Scientific, USA), 4 mM L-glutamine, 10 U/mL penicillin, and 0.1 mg/mL streptomycin (Sigma Aldrich, USA), at 34°C in an atmosphere without CO2. BHK-21 cells, obtained from the ATCC, were maintained in D-MEM (Sigma Aldrich, USA), supplemented with 10% v/v FBS, 4 mM L-glutamine, 10 CFU/mL penicillin, and 0.1 mg/mL streptomycin at 37°C with 5% CO2, and used for plaque assays. Conjugated antibodies against CD14 (clone M5E2), TLR3 (clone TLR3.7), TLR4 (clone HTA 125), and TLR9 (clone eB72-1665) were purchased from eBioscience (USA).

### Blood samples from healthy donors

Venous peripheral blood samples were obtained from healthy individuals aged 20–40 years who had not been previously vaccinated against yellow fever virus and were seronegative for the DENV NS1 antigen and DENV IgM/IgG, as determined by the SD BIOLINE Dengue Duo rapid test (Standard Diagnostics). All our experiments were performed with cells from at least six healthy donors.

### Monocyte isolation and monocyte-derived macrophage differentiation (MDMs)

To obtain MDMs, peripheral blood mononuclear cells (PBMCs) were obtained from 50 mL of peripheral blood from healthy individuals with 2% v/v ethylenediaminetetraacetic acid, as

described previously [17,18]. Briefly, the PBMCs were separated using density gradient centrifugation with Lymphoprep (STEMCELL technologies, USA) at $800 \times g$ at room temperature for 20 min and then washed three times with phosphate-buffered saline (PBS) at $250 \times g$ for platelet removal. The PBMCs were suspended in RPMI-1640 medium (Sigma Aldrich, USA) supplemented with 0.5% autologous heat-inactivated serum (30 min at 56°C). Monocytes were then obtained from the PBMCs by plastic adherence, as described previously [17]. Briefly, $1 \times 10^5$ cells were stained with anti-CD14-FITC (clone M5E2, BD Biosciences, USA) and analyzed by flow cytometry. The percentage of CD14+ cells was used to seed $5 \times 10^5$ CD14+ cells into 24-well plates (Corning Incorporated Life Sciences, USA) in RPMI-1640 medium supplemented with 0.5% inactivated autologous serum and cultured at 37°C with 5% $CO2$ to allow enrichment of monocytes through plastic adherence. After 3 h of adherence, non-adherent cells were removed by extensive washing with pre-warmed PBS supplemented with 0.5% FBS. Adherent cells were then cultured in RPMI-1640 medium supplemented with 10% FBS at 37°C with 5% $CO2$ for 6 days to obtain MDMs. Fresh medium with 10% FBS was replenished every 48 h. The purity of MDMs and $D_3$-MDMs was evaluated by measuring the presence of contaminant cell populations, including CD19+, CD3+, and CD56+ in monocytes before differentiation, and measuring CD68+ cells after 6 days of differentiation.

## $D_3$-MDMs differentiation in the presence of Vitamin D3

Monocytes were differentiated for 6 days in the presence of 1α,25-dihydroxyvitamin D3 (VitD3; Sigma Aldrich, USA), at a concentration of 0.1 nM which represents the physiological and therapeutical concentration [23,24], and as we have previously described [17,18]. VitD3 was replenished with fresh medium every 48 h. To evaluate the effect of VitD3 on $D_3$-MDMs cultures, vitamin D receptor (VDR) and cytochrome P450 family 24 subfamily A member 1 (CYP24A1) gene expression were evaluated using quantitative polymerase chain reaction (qPCR). The biological activity VitD3 was determined by the transcriptional induction of VitD3 signaling targets, such as VDR and CYP24A1 [17]. The purity of $D_3$-MDMs was repeatedly above 90%, as measured by the presence of contaminant cell populations, including CD19+, CD3+, and CD56+ (non-myeloid cells).

## Virus stocks and titration

DENV-2 New Guinea C was provided by the Centers for Disease Control and Prevention (CDC, USA). Viral stocks were obtained by inoculating a monolayer of C6/36 HT cells in a 75-cm2 tissue culture flask with the virus at a multiplicity of infection (MOI) of 0.05 diluted in L-15 supplemented with 2% FBS. After 3 h of adsorption, fresh L-15 medium supplemented with 2% FBS was added, and the cells were cultured for 5 days at 34°C without $CO2$. The supernatant was obtained by centrifugation at $1000 \times g$ for 5 min to remove cellular debris and then aliquoted and stored at $-70$°C until use. Virus titration was performed by quantification of plaque-forming units (PFU) using a plaque assay. A total of $5 \times 10^4$ BHK-21 cells/well were cultured in 24-well plates in D-MEM supplemented with 2% v/v FBS, 4 mM L-glutamine, 10 U/mL penicillin, and 0.1 mg/mL streptomycin and incubated at 37°C with 5% $CO2$ overnight. Next, cells were infected with 10-fold serial dilutions of the virus in 250 μL of the medium. After 2 h of adsorption, the virus was removed, washed once with PBS and plaque medium of D-MEM containing 1.5% m/v carboxymethylcellulose sodium salt (medium viscosity, Sigma-Aldrich, USA), 3% NaCO3, and 1% HEPES (Sigma-Aldrich, USA), and 10 CFU/mL penicillin, 0.1 mg/mL streptomycin, and 1% of FBS were added to the cells. Then, the cells were incubated at 37°C with 5% $CO2$ for 5 days. Next, the plaque medium was removed, and the cells were washed twice with PBS and stained with 4% m/v crystal violet solution and 3.5% v/v

formaldehyde (Merck, Germany) for 30 min. After staining, cells were washed once with PBS, and the plaque count was performed manually to obtain PFU/mL.

## MDMs and D₃-MDMs infection with DENV

Both MDMs and $D_3$-MDMs monolayers were challenged with DENV-2 at an MOI of 5, diluted in 300 μl of RPMI-1640 medium supplemented with 2% FBS. Two hours post-infection (hpi), cells were washed with PBS, and the medium was replenished with RPMI 10% FBS and then cultured at 37˚C 5% CO2. At 2, 8, and 24 hpi, monolayers were harvested, and either the percentage of infection was determined by flow cytometry, or total RNA extraction was carried out and used for viral RNA quantification, while the supernatants were used for viral titration by plaque assay and for quantification of cytokine production.

## Flow cytometry assays

Flow cytometry was used to assess the frequency of DENV-infected cells, expression levels of VDR and, changes of TLR3/4/9 expression, as previously described [17,18]. Flow cytometry was also used for assessing the production of ROS production by the detection of the dihydror-hodamine 123 (DHR 123; Sigma Aldrich, USA). DENV infection was evaluated through the intracellular detection of DENV E antigen at the indicated time points and the cells were fixed using a fixation/permeabilization buffer (eBioscience, USA). Following washing steps with PBS, cells were stained with the monoclonal antibody, 4G2 (Millipore, Germany) for 40 min, followed by 40 min staining with goat anti-mouse IgG-FITC (Thermo Scientific, USA). Expression of TLR3, TLR4, and TLR9 in DENV-2-infected and mock-infected cells was evaluated at 2, 8, and 24 hpi as we described in [25]. All data acquisition and analysis were done using the BD FACScan system and FACSDiva software, respectively.

## Quantitation of viral RNA copy number

Total RNA was purified from DENV-2-infected and mock-infected MDMs and D₃-MDMs using TRIzol reagent (Thermo Scientific, USA) following the manufacturer's instructions. The RNA concentration was quantified using a NanoDrop spectrophotometer (NanoDrop Technologies, USA). Then, cDNA was synthesized using random primers from a standard concentration of 50 ng of RNA and the RevertAid H Minus First Strand cDNA Synthesis Kit (Thermo Scientific, USA) following the manufacturer's instructions. Viral RNA copy quantification with cDNA was carried out using the specific primers depicted in S1 Table, as described previously [26]. qPCR was performed with Maxima SYBR Green qPCR Master Mix (Thermo Scientific, USA) and analyzed with the CFX96 Touch Real-Time PCR Detection System (Bio-Rad, USA). The calculation of viral RNA copies was based on a standard curve of Ct values of 10-fold serial dilutions of a plasmid encoding the full genome of DENV-2 of known length and concentration, as previously described [27].

## Cytokine production

The levels of IL 6, TNF-α, and IL 10 were assessed in supernatants from MDMs and D₃-MDMs infected with DENV-2 at 2, 8, and 24 hpi, using an ELISA assay (BD OptEIA, BD Biosciences, USA), following the manufacturer's recommendations.

## Quantification of gene expression

The mRNA quantification of TLR3, TLR4, TLR7, TLR8, TLR9, RIG I IFN-α, IFN-β, protein kinase R (PKR), 2′-5′-oligoadenylate synthetase 1 (OAS1), VDR, CYP24A1, SOCS-1, and

ubiquitin was performed in DENV-2-infected and mock-infected MDMs and D3-MDMs using qPCR. Briefly, cDNA was synthesized from total RNA using random primers, a standard concentration of 50 ng of RNA, and the RevertAid H Minus First Strand cDNA Synthesis Kit (Thermo Scientific, USA). Then, qPCR was performed using Maxima SYBR Green (Thermo Scientific, USA), following the manufacturer's instructions with specific primers for each gene (S1 Table). The specificity of the amplification product was determined by melting curve analysis. The relative quantification of each mRNA was normalized to the constitutive gene, ubiquitin, and mock-treated MDMs and D3-MDMs from each time point evaluated (e.g. 2 hours mock MDM vs 2 hours DENV-2-infected MDMs), using the $\Delta\Delta Ct$ method and reported as the fold change.

### Detection of ROS production

For quantification of ROS production, MDMs and $D_3$-MDMs were infected with DENV-2 for 2, 8, and 24 h, or stimulated with zymosan at a final concentration of 10 µg/mL for 4 h. Thereafter, cells were stained with DHR 123 (Sigma Aldrich, USA) at a final concentration of 3.5 µg/mL for 1 h at 37°C in the dark. Then, the analysis was performed by flow cytometry using a FACScan flow cytometer and the FACSDiva software. The production of ROS was inhibited by treatment with N-acetylcysteine (NAC) for 1 h at a final concentration of 10 mM. Treated and non-treated cells were then infected with DENV-2 or treated with zymosan. ROS production and TLR9 expression were measured as described above.

### Statistical analysis

Comparisons between MDMs and $D_3$-MDMs were undertaken using a matched Wilcoxon and two-way analysis of variance (ANOVA) along with Bonferroni posthoc tests, depending on the data analyzed A value of $p < 0.05$ was considered statistically significant. The calculation of these parameters was carried out using GraphPad Prism version 6 (GraphPad Software, USA) software.

## Results

### VitD3 modulates the expression of RIG I, TLR7, TLR3 and TLR4 in $D_3$-MDMs infected with DENV-2

We have previously shown that differentiation of MDMs in the presence of VitD3 confers partial resistance to DENV-2 infection, due to a decrease in the expression of MR, accompanied by a reduced inflammatory response of $D_3$-MDMs [17]. Here, we sought to test if VitD3 differentially regulates innate immune responses in MDMs by modulating the expression of PRRs known to sense DENV infection. With this in mind, first, we differentiate the MDMs in the presence or absence of vitamin D and determine the purity. We found that the purity of MDMs and $D_3$-MDMs was repeatedly above 90% (S1A–S1D Fig). Next, we confirmed that VitD3 decreases the percentage of DENV-infected $D_3$-MDMs (S2A and S2B Fig) as we previously reported [17]. In addition, we showed that VitD3 significantly decreased DENV replication according to the viral RNA copy number and viral load, determined by qPCR and plaque assay, respectively (S2C and S2D Fig). Further, as we previously reported [17,18], lower levels of IL-6 and TNF-α were found in DENV-2-infected $D_3$-MDMs at 24 hpi, compared with DENV-2-infected MDMs, confirming a reduced inflammatory response in those cells (Figs S3A and 3B).

 In the current study we quantified the mRNA expression of RIG-I, TLR7, TLR8, and the mRNA and protein level of TLR3 and TLR4 in response to DENV-2 infection, in $D_3$-MDMs

and MDMs infected at 2, 8, and 24 h. The expression mRNA levels of the tested genes did not change over time in mock-treated MDMs and D3-MDMs, at 2, 8, and 24 h (S4 Fig), and these results were used to normalize the data from DENV-2 infected samples. The successful effect of VitD3 in the differentiation of D$_3$-MDMs was confirmed by the increase in the transcription of the target gene CYP24A1 (S5A Fig), and VitD3 receptor (VDR) at the protein level (S5B and S5C Fig), in both DENV-2 infected and mock-infected cells.

DENV-2 infection in both MDMs and D$_3$-MDMs led to an increase in RIG-I mRNA expression, compared to mock-treated cells, with the highest expression measured at 24 hpi; although at this time point, we observed no significant differences in RIG-I mRNA expression between MDMs and D$_3$-MDMs (Fig 1A). However, a significant decrease in the induction of RIG-I mRNA expression was noted at 8 hpi in D$_3$-MDMs as compared to MDMs (Fig 1A). Expression levels of TLR7 mRNA remained at baseline levels at 2 and 8 hpi in both MDMs and D$_3$-MDMs infected, whereas a 3-fold increase in TLR7 mRNA expression occurred only in infected-MDMs at 24 hpi and was not observed in infected-D$_3$-MDMs (Fig 1B). Expression levels of TLR8 mRNA did not change among DENV-2 infected MDMs and D$_3$-MDMs (S5D Fig). On the other hand, the expression of TLR3 mRNA was significantly downregulated in D$_3$-MDMs at 8 and 24 hpi compared with MDMs in response to DENV-2 infection (Fig 1C). In the same way, a significant decrease in TLR3 protein levels in D$_3$-MDMs compared with MDMs at 2 and 8 hpi (Figs 1D and S5E) was observed, suggesting early downregulation of TLR3 expression in infected-D$_3$-MDM. The highest expression of TLR4 mRNA was observed at 8 hpi in both D$_3$-MDMs and MDMs, although no significant differences were observed (Fig 1E). However, we did observe a significant increase in TLR4 mRNA expression at 24 hpi in DENV-2-infected D$_3$-MDMs. Likewise, a significant increase in TLR4 protein level was found in D$_3$-MDMs compared with MDMs at 24 hpi (Figs 1F and S5F). At early time points of infection (2 and 8 h), VitD3 did not affect TLR4 expression.

To test if the observed regulation of TLRs was related to a direct effect of VitD3 rather than restricted infection of DENV-2 in D$_3$-MDMs, we infected MDMs and D$_3$-MDMs with different MOIs of DENV-2. As shown in Fig 2A, VitD3 decreased the percentage of infected cells, regardless of the MOI used. An additional analysis was done to confirm these results, and the % of suppression of infection was calculated with each MOI (Fig 2B). We observed that suppression of infection induced by VitD3 was similar among different MOIs, except for MOI of 1 vs MOI of 7, which appeared to be the conditions in which VitD3 had the highest and the lowest effect in DENV-2 restriction, respectively (Fig 2B). Then, we compared the expression of TLR3 and TLR7 mRNA in MDMs and D$_3$-MDMs infected with different DENV-2 MOIs and observed that VitD3 regulated the expression of these TLRs with several of the MOI used (Fig 3A and 3B). Notably, when we stratified the data based on the matched percentage of infected cells (e.g., MDMs MOI 3 vs. D$_3$-MDMs MOI 5), D$_3$-MDMs still showed a significant decrease in TLRs mRNA expression induced by VitD3 (Fig 3D and 3E). These findings suggest that regulation of TLR3 and TLR7 expression by VitD3 in D$_3$-MDMs is a result of the direct effect of the treatment rather than a consequence of reduced levels of viral replication. We suggest that the downregulation of TLR expression by VitD3 could be contributing to the regulation of the inflammatory response, in addition to a decreased C-type lectin activation that was previously reported by us in [17].

## VitD3 decreases the expression of TLR9 and the production of ROS in D$_3$-MDMs infected with DENV-2

TLR9 is known to recognize unmethylated CpG motifs in DNA and cannot recognize viral RNA components. However, previously we reported that myeloid DCs of dengue infected

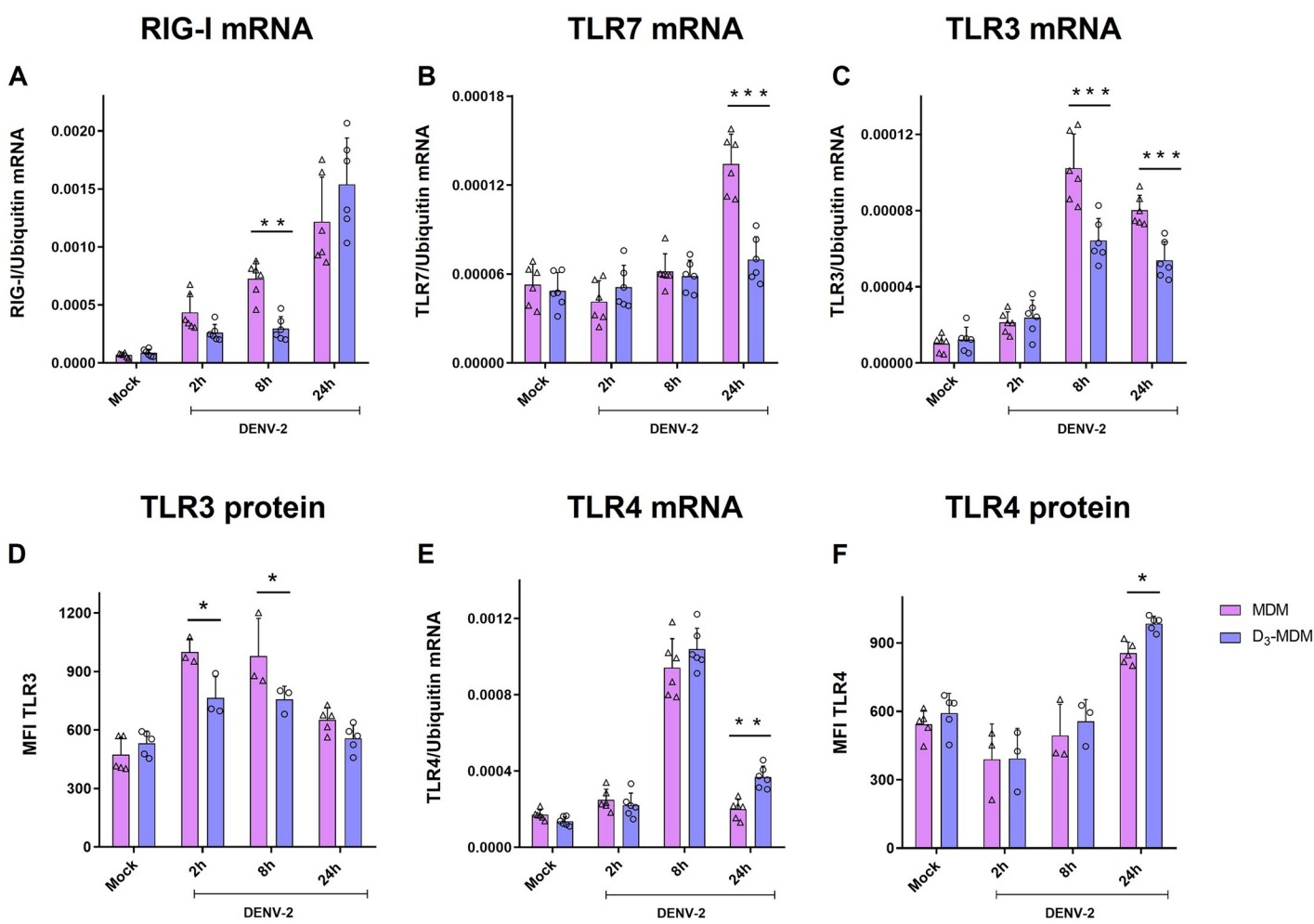

**Fig 1. VitD3 modulates the expression of TLRs in D₃-MDMs.** The MDMs were differentiated in the presence of VitD3 (0.1 nM) for 6 days (D₃-MDMs) and then infected with DENV-2 with an MOI of 5 for 2, 8, or 24 h. Expression of RIG I (A), TLR7 (B), TLR3 (C), and TLR4 (E) was measured by qPCR in MDMs and D₃-MDMs using the gene encoding ubiquitin as a housekeeper gene. Data are expressed as fold change relative to mock-treated MDMs and D₃-MDMs from each time point. Expression of TLR3 (D) and TLR4 (F) protein was measured by flow cytometry and expressed as mean fluorescence intensity. Six different donors were analyzed. Differences were identified using a two-way ANOVA with a Bonferroni posthoc test and a 95% confidence interval (***p < 0.001, **p < 0.01, *p < 0.05).

patients have an increased expression of TLR9 compared to healthy controls [25]. In addition, upregulation and activation of TLR9 in DENV-2-infected Mo-DCs were described recently [28]. In line with this study, we found an upregulation of TLR9 mRNA in MDMs after DENV-2 infection at 8 and 24 hpi (Fig 4A), and an increase in TLR9 protein expression at 2, 8, and 24 hpi quantified by flow cytometry (Fig 4B). Interestingly, we observed significant downregulation in TLR9 mRNA and protein expression in D₃-MDMs at 8 and 24 hpi (Figs 4A, 4B and S5G). We also compared the expression levels of TLR9 mRNA in matched DENV-2-infected MDMs and D₃-MDMs, as described above, and found that the decrease in the expression of this TLR was a direct effect of VitD3 (Fig 3C–3F), suggesting that VitD3 may counteract the effect of DENV-2 induction on TLR9 expression.

Considering that the upregulation of TLR9 expression in Mo-DCs was dependent on ROS production [28], the production of total ROS in MDMs and D₃-MDMs after DENV-2 infection was evaluated at 2, 8, and 24 hpi by staining with DHR 123. At 2 hpi a significant increase in the production of ROS by both MDMs and D₃-MDMs in response to DENV-2 infection

   

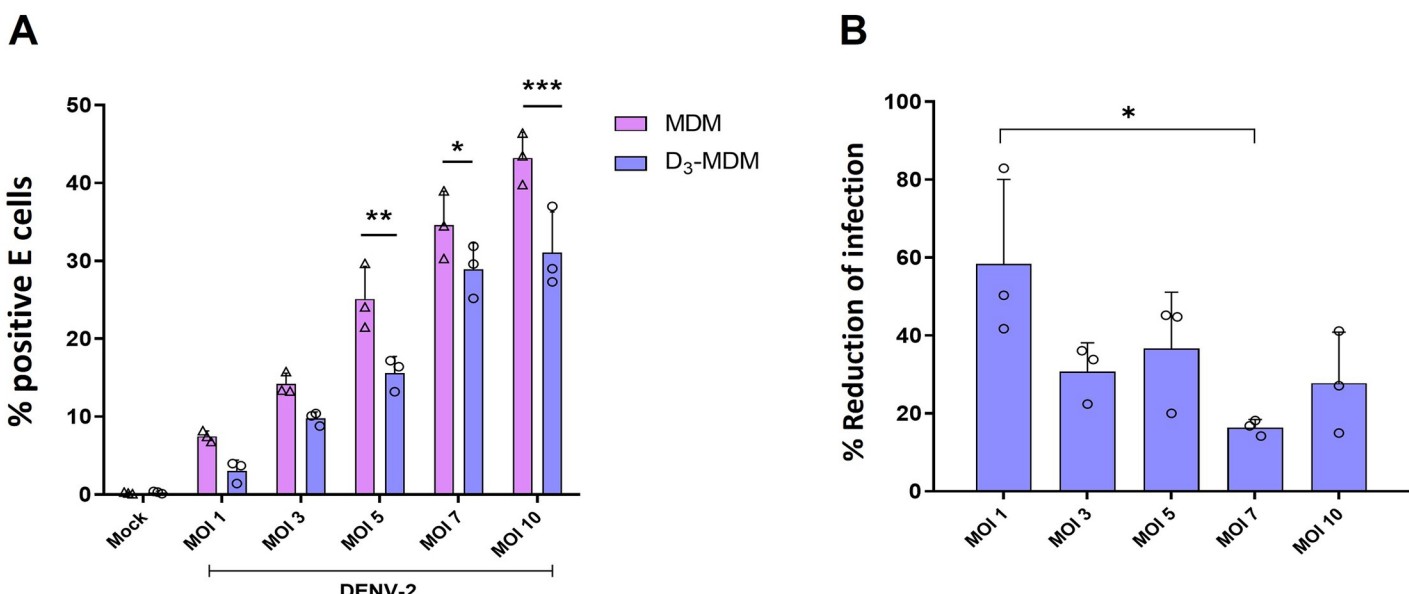

**Fig 2. Reduction of E-positive cells in D$_3$-MDMs infected with different MOI of DENV-2.** The MDMs were differentiated in the presence of VitD3 (0.1 nM) for 6 days (D$_3$-MDMs) and then infected with DENV-2 with different MOI for 24 h. Infection was evaluated through flow cytometry and expressed as % E-positive cells. Three different donors were analyzed. Differences were obtained with a two-way ANOVA with a Bonferroni posthoc test and a 95% confidence interval (***p < 0.001, **p < 0.01, *p < 0.05).

was observed, yet levels produced by MDMs were significantly higher than those observed in D$_3$-MDMs (Fig 4C). At later time points, the production of ROS in DENV-2-infected MDMs and D$_3$-MDMs was restored to the levels detected in mock-infected cells (Fig 4C). To test whether a direct relationship exists between ROS production and the upregulation of TLR9 mRNA, we proceeded to inhibit the production of ROS, using the ROS scavenger, NAC. Using 10 mM of NAC, a significantly decrease ROS production in MDMs treated with zymosan, used as a positive control, was observed, and also in DENV-2 infected MDMs at 2 hpi, obtaining a level of ROS similar to mock-infected cells (Fig 4D). Interestingly, when TLR9 mRNA expression was quantified in MDMs infected with DENV-2 and treated with 10 mM of NAC, a significant decrease in TLR9 was observed at all time points evaluated, compared with non-treated and DENV-2-infected MDMs (Fig 4E).

Altogether, these results suggest that DENV-2 infection induces early production of ROS in MDMs that, in turn, upregulates the expression of TLR9. Conversely, VitD3 downregulates both the early production of ROS in DENV-2-infected D$_3$-MDMs and both mRNA and protein levels of TLR9.

### VitD3 upregulates mRNA expression of SOCS-1, PKR, and OAS1 in DENV-2-infected D$_3$-MDMs

SOCS family of proteins are important in the regulation of the inflammatory response upon activation of TLRs [22]. Thus, the expression of SOCS-1 mRNA in DENV-2-infected D$_3$-MDMs and MDMs was compared at 2, 8, and 24 hpi. As shown in Fig 5A, at 2 hpi, there were no differences in expression levels of SOCS-1 mRNA between D$_3$-MDMs and MDMs; however, we observed an increase in SOCS-1 in DENV-2-infected D$_3$-MDMs at 8 and 24 hpi compared with MDMs.

Finally, regulation of other antiviral mechanisms by VitD3, such as regulation of the IFN-I response, was explored. Previously, it was reported that VitD3 can enhance the activity of

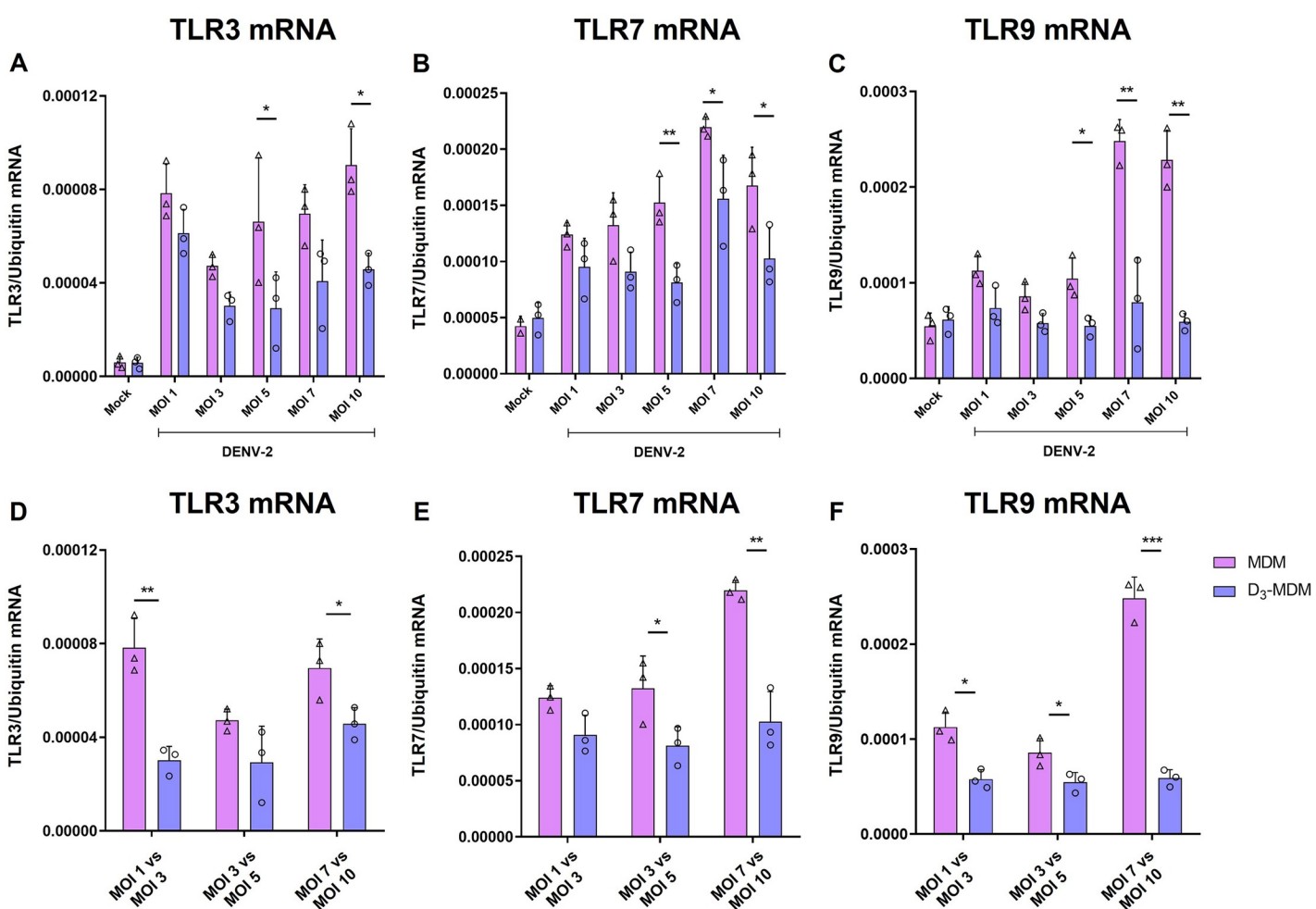

**Fig 3. Regulation of TLR expression is independent of the MOI used.** The MDMs were differentiated in the presence of VitD3 (0.1 nM) for 6 days (D$_3$-MDMs) and then infected with DENV-2 with different MOI for 24 h. Expression of TLR3 (A and D), TLR7 (B and E), and TLR9 (C and F) were measured by qPCR in MDMs and D$_3$-MDMs using ubiquitin as housekeeping. Data are expressed as fold change relative to mock-treated MDMs and D$_3$-MDMs. Three different donors were analyzed. Differences were obtained with a two-way ANOVA with a Bonferroni posthoc test and a 95% confidence interval (***$p < 0.001$, **$p < 0.01$, *$p < 0.05$).

IFN-I, which was evidenced by an increase in ISG expression [29]. The mRNA levels of IFN-α, IFN-β, PKR, and OAS1 were quantified in D$_3$-MDMs and MDMs infected with DENV-2 at 2, 8, and 24 hpi. The expression of tested genes did not change over time in mock-treated D$_3$-MDMs (S2 Fig). There was a significant increase in the expression of both IFN-α and IFN-β mRNA in MDMs in response to DENV-2 infection, especially at 24 hpi (Fig 5B and 5C). However, IFN-α mRNA expression was downregulated in DENV-2-infected D$_3$-MDMs at 24 hpi compared with MDMs (Fig 5B), possibly due to lower levels of viral replication in D$_3$-MDMs. Conversely, there was no difference in IFN-β mRNA expression in D$_3$-MDMs compared with MDMs infected with DENV-2 (Fig 5C). Although D$_3$-MDMs showed a lower expression of IFN-α and no increase in IFN-β, a significantly higher expression of both PKR and OAS1 mRNA was found in DENV-2-infected D$_3$-MDMs compared with MDMs at 24 hpi (Fig 5D and 5E).

To check if the noted differences between MDMs and D$_3$-MDMs were due to different levels of infection in these cells, we evaluated the expression of SOCS-1, PKR, and OAS1 mRNA in MDMs and D$_3$-MDMs infected with different DENV-2 MOIs, as described in Fig 3. As

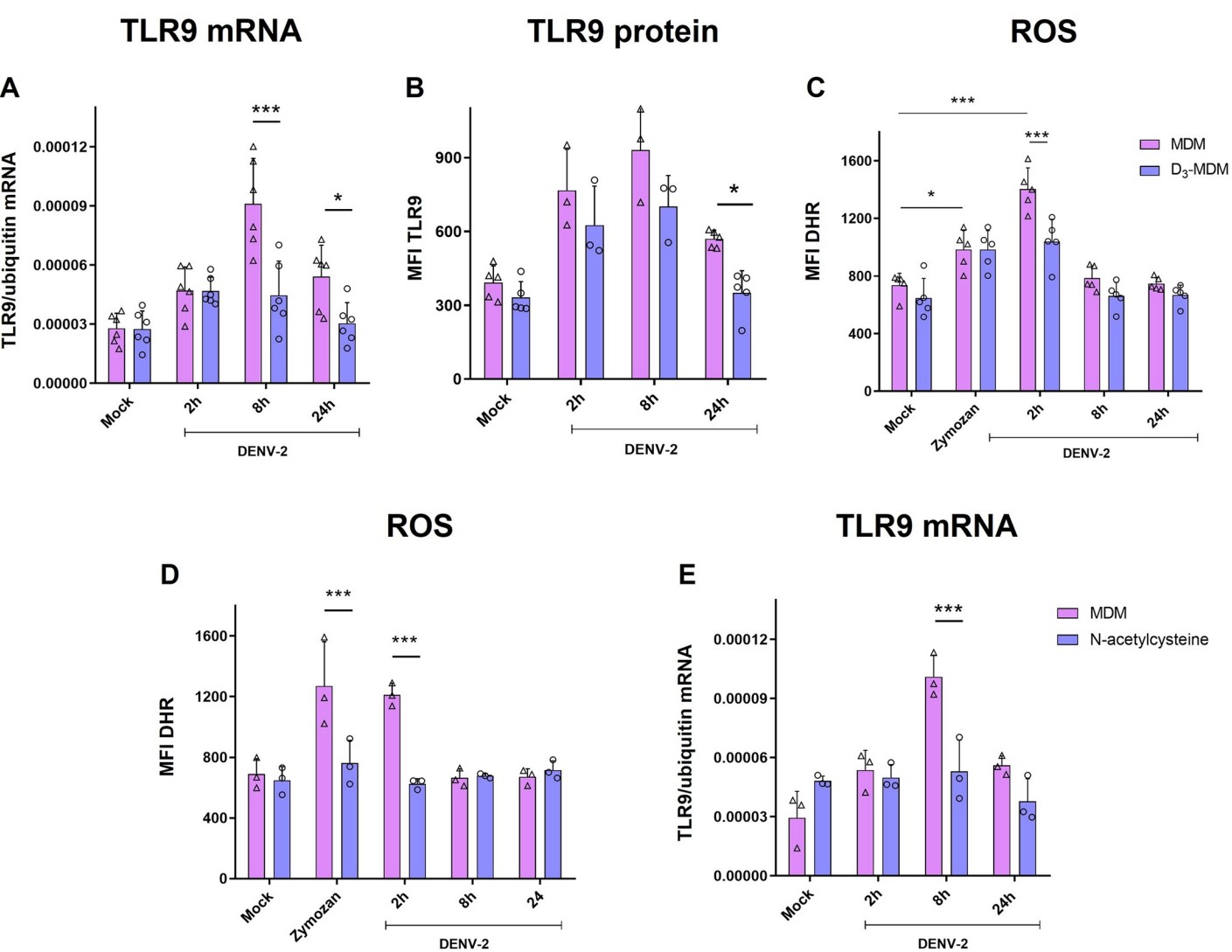

**Fig 4. VitD3 downregulates TLR9 expression and ROS production in D$_3$-MDMs.** MDMs were differentiated in the presence of VitD3 (0.1 nM) for 6 days (D$_3$-MDMs) and then infected with DENV-2 with an MOI of 5 for 2, 8, or 24 h. Expression of TLR9 (A) was measured by qPCR in MDMs and D$_3$-MDMs using the gene encoding ubiquitin as a housekeeper gene. Data are expressed as fold change relative to mock-treated MDMs and D$_3$-MDMs from each time point. Expression at the protein level of TLR9 (B) was measured by flow cytometry and expressed as mean fluorescence intensity. The production of ROS was measured by flow cytometry through DHR 123 in MDMs and D$_3$-MDMs infected with DENV-2 with an MOI of 5 for 2, 8, and 24 h, or stimulated with 10 μg/mL of zymosan (C). The MDMs were pre-treated with N-acetylcysteine at 10 mM for 1 h and then infected with DENV-2 with an MOI of 5 for 2, 8, and 24 h or treated with 10 μg/mL of zymosan. Production of ROS was measured by flow cytometry through DHR 123) (D) and the TLR9 expression was measured by qPCR (E). Six different donors (A, B, and C) or three different donors (D and E) were analyzed. Differences were obtained with a two-way ANOVA with a Bonferroni posthoc test and a 95% confidence interval ($^{***}p < 0.001$, $^{**}p < 0.01$, $^{*}p < 0.05$).

shown in Fig 6A, 6B and 6C, regulation of SOCS-1, PKR, and OAS1 mRNA expression induced by VitD3 were MOI-independent. Also, the expression levels of SOCS-1, PKR, and OAS1 mRNA were higher in D$_3$-MDMs than in MDMs when matching infection rate conditions were done (e.g., MDMs MOI 3 vs. D$_3$-MDMs MOI, Fig 6D, 6E and 6F), suggesting a direct effect of VitD3 on the regulation of these genes in D$_3$-MDMs. Taken together, our results may suggest that VitD3 can increase the expression of ISGs such as PKR and OAS1 in D$_3$-MDMs after DENV-2 infection, which could constitute an additional antiviral mechanism of VitD3 in addition to the downregulation of MR reported by us previously [17].

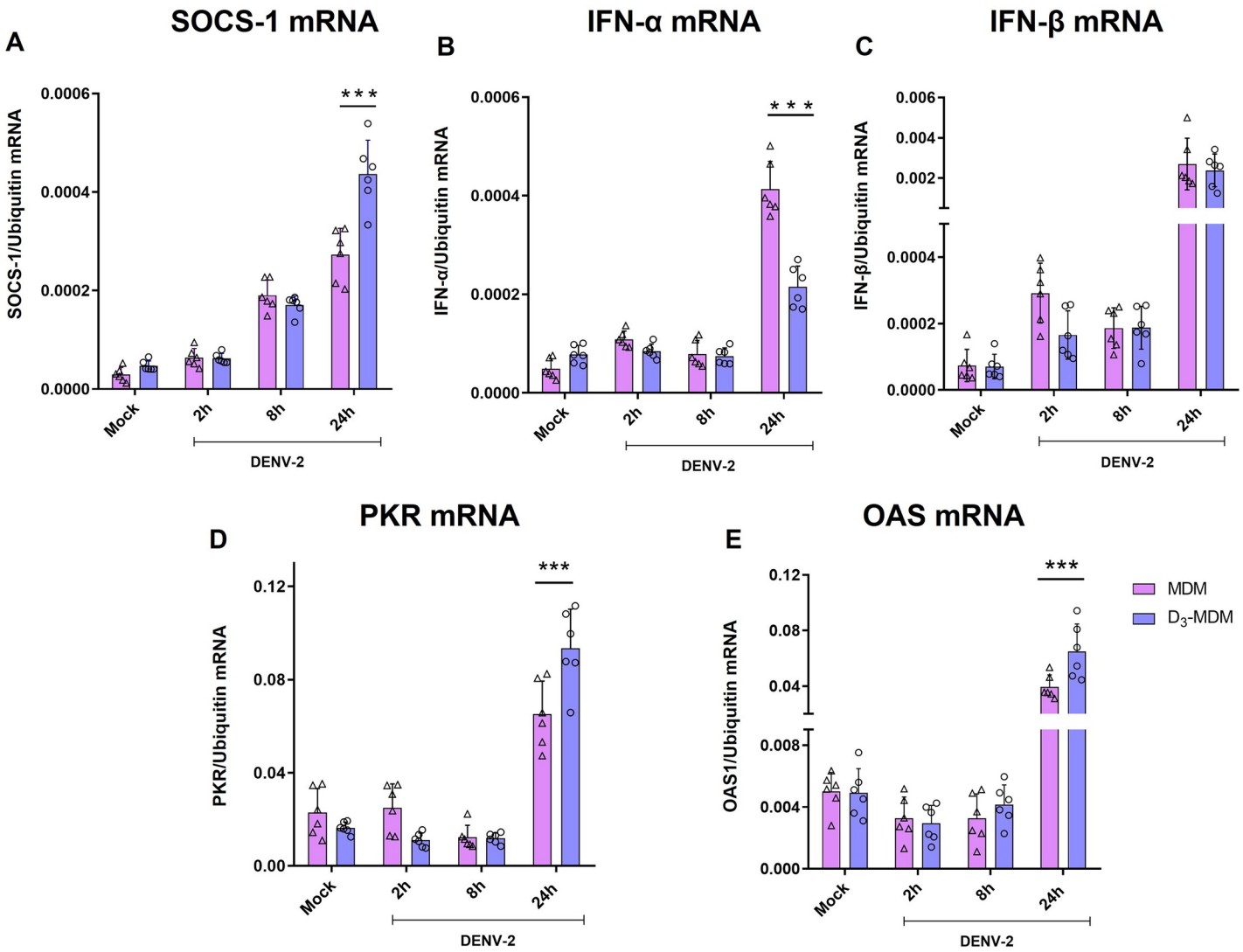

**Fig 5. VitD3 upregulates the expression of SOCS-1 and ISGs in D$_3$-MDMs.** The MDMs were differentiated in the presence of VitD3 (0.1 nM) for 6 days (D$_3$-MDMs) and then infected with DENV-2 for 2, 8, or 24 h. Expression of SOCS-1 (A), IFN α (B), IFN β (C), PKR (D), and OAS1 (E) were measured by qPCR in MDMs and D$_3$-MDMs using the gene encoding ubiquitin as a housekeeper gene. Data are expressed as fold change relative to mock-treated MDMs and D$_3$-MDMs from each time point. Six different donors were analyzed. Differences were obtained with a two-way ANOVA with a Bonferroni post-hoc test and a 95% confidence interval (***$p < 0.001$, **$p < 0.01$).

## Discussion

Compounds exhibiting both anti-inflammatory and antiviral properties responses are considered promising therapeutic options to prevent severe dengue. In recent years, VitD3 has drawn the attention of the field due to its wide antiviral and immunomodulatory properties [14,16,30,31]. Indeed, we have recently shown that D$_3$-MDMs express lower levels of MR, which conferred partial resistance to DENV-2 infection [17]. In the present study, we observed that the presence of VitD3 during the differentiation of macrophages downregulated the expression of major PRRs and inflammatory pathways while increasing the expression of ISGs (Fig 7).

Here we observed that VitD3 significantly reduced the expression of TLR3, TLR7, and TLR9 in DENV-2-infected D$_3$-MDMs. However, different from TLR3 and TLR9, reduced

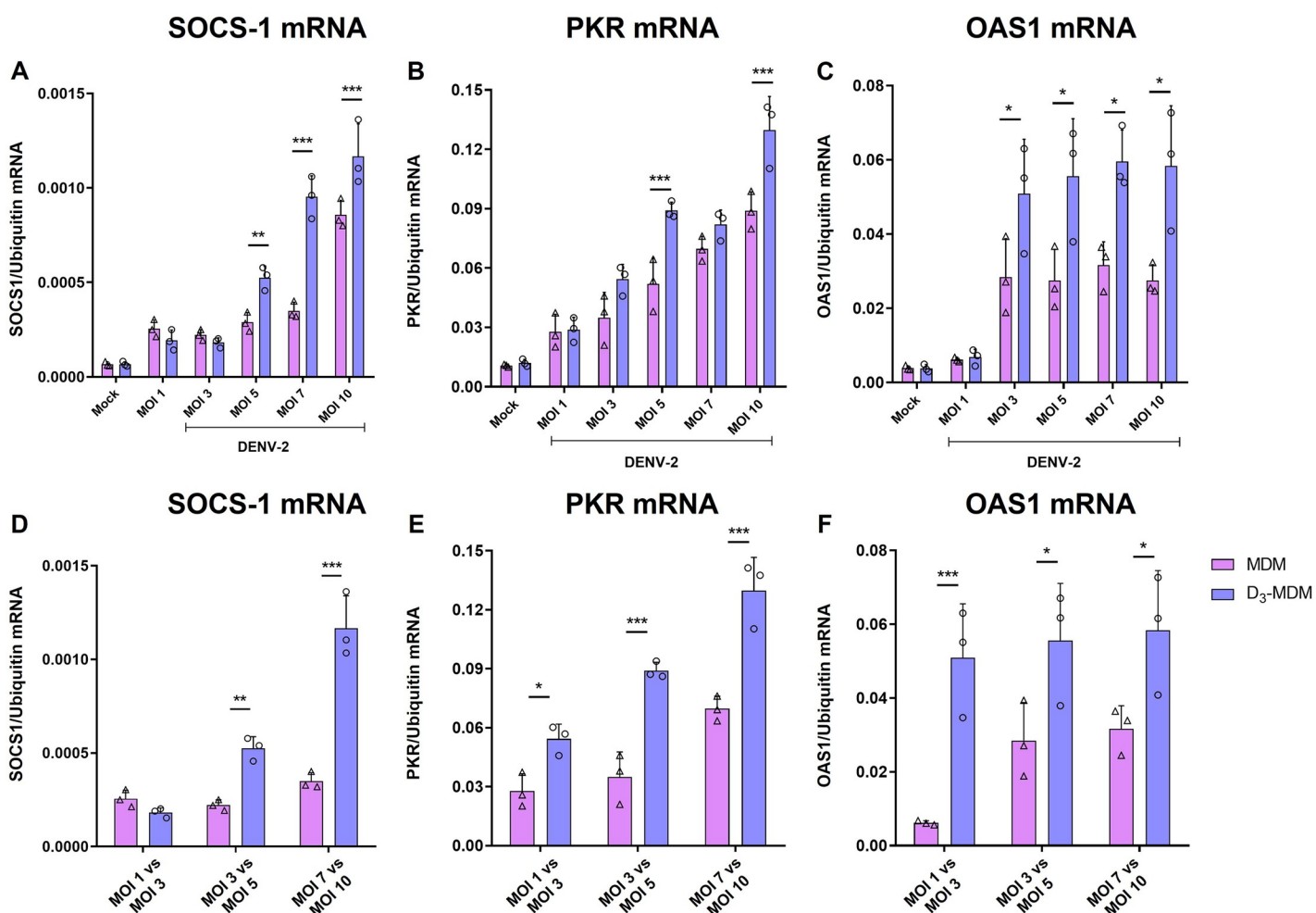

**Fig 6. Regulation of SOCS-1 and ISGs expression is independent of the MOI used.** The MDMs were differentiated in the presence of VitD3 (0.1 nM) for 6 days (D$_3$-MDMs) and then infected with DENV-2 with different MOIs for 24 h. Expression of SOCS-1 (A and D), PKR (B and E), and OAS1 (C and F) were measured by qPCR in MDMs and D$_3$-MDMs using ubiquitin as a housekeeper. Data are expressed as fold change relative to mock-treated MDMs and D$_3$-MDMs. Three different donors were analyzed. Differences were obtained with a two-way ANOVA with a Bonferroni posthoc test and a 95% confidence interval (***$p < 0.001$, **$p < 0.01$, *$p < 0.05$).

expression of TLR7 was not be confirmed at the protein level, which is a limitation of our study. Regardless, regulation of TLRs was observed at both early and late time points of infection, suggesting that VitD3-mediated responses counteract the effect of DENV infection on TLR expression [32,33]. To note, we found discrepancies between the kinetics of mRNA and protein expression of TLR3, noticing a faster increase in protein expression than mRNA levels. Unfortunately, these findings were not dissected in detail in this study and potential explanatory mechanisms remain speculative. For example, transcriptome and proteomic analyses indicate that mRNA abundance does not always correlate with protein levels [34,35]. In the same way, some studies have suggested that is not easy to predict protein expression levels from quantitative mRNA data [35]. Furthermore, in stress conditions, such as DENV infection [36,37], changes in tRNA abundance can affect protein expression independently of mRNA abundance [38]. Finally, pre-formed TLR3 complexes could be recruited to endosomes rapidly after DENV recognition, similar to TLR2 recruitment to lipid rafts after recognition of other types of ligands [39]. Further studies would be important for further evaluation of these hypotheses.

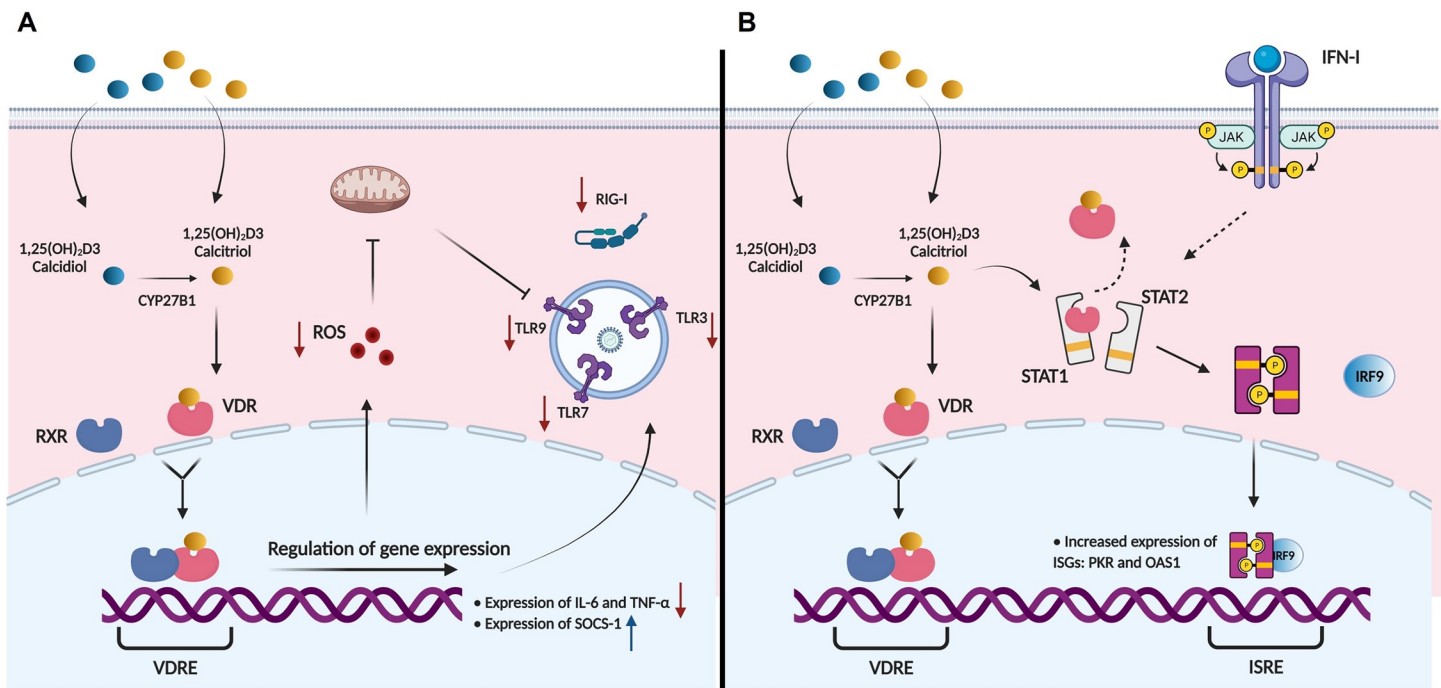

**Fig 7. Hypothetical model of VitD3 effects in $D_3$-MDMs during DENV-2 infection.** (A) VitD3 can be found as inactive form 25(OH)₂D2 (calcidiol), as active form 1,25 (OH)₂D3 (calcitriol), or be converted from inactive to active form within the cell by the action of CYP7B1. VitD3 interacts with the vitamin D receptor (VDR) present in the nuclei and associates then, with the nuclear receptor retinoid X receptor (RXR). Together, they act as transcription factors that will increase the expression of genes that have vitamin D response elements (VDRE) [58,59]. Genomic effects of VitD3 during DENV-2 infection include decreased expression of IL-6, TNF-α, RIG-I, TLR3, TLR7 and decreased production of reactive oxygen species (ROS). Since ROS can induce mitochondrial damage and then increase activation and expression of TLR9 [28], VitD3 reduced TLR9 expression indirectly. (B) As a non-genomic effect, VitD3 can bind to VDR that is initially bound to STAT1 [29], and indirectly enhance the activity of the JAK-STAT signaling pathway and IFN-I activity, thus increasing expression of PKR and OAS1.

Other studies have shown that VitD3 modulates the expression of various TLRs as we observed in our study. For instance, PBMCs obtained from patients with systemic lupus erythematosus that were treated with 50 nM of VitD3 express lower levels of TLR3, TLR7, and TLR9 compared to non-treated PBMCs [40], as observed in this study. Culturing human corneal epithelial cells with 100 nM of VitD3 for 24h leads to a reduced expression of TLR3 along with a reduced response to poly-IC stimulation [41]. On the other hand, in a recent study in which DENV-2-infected U937 cells were treated with 0.1 or 1 μM of VitD3, no effect on TLR3 expression was observed [15]. The observed differences could be due to different cell types and concentrations of VitD3; however, overall, most studies show downregulation of TLR3, TLR7, and TLR9 by VitD3 like we found. Whether regulation of TLR3 by VitD3 is dose-dependent, or if different baseline levels of VitD3 in DENV- infected patients are important for host response and control of infection remains to be studied.

Some TLRs have been implicated in DENV pathogenesis. For instance, recognition of DENV infection by TLR4 and TLR2 has been shown to mediate endothelial activation and permeability [42,43]. Increased TLR2/4 expression was also observed on monocytes of acute infected patients [44]. While we did not measure TLR2 expression and it is a limitation of our study, we found that DENV-2 infection led to an upregulation of TLR4 in $D_3$-MDMs and MDMs. Notably, however, the increase was significantly higher in $D_3$-MDMs than in MDMs. This was unexpected, as VitD3 has been shown to downregulate the expression of TLR4 in monocytes [20], keratinocytes [45], and PBMCs [46], which is accompanied by a decreased response to stimulation with LPS. However, the authors in these studies used higher

concentrations of VitD3 (ranging from 0.1–1 μM) than we used (0.1 nM), and these high concentrations are not within the conventional and safe treatment in humans. Although the overall effect of VitD3 on TLR4 expression is likely to be cell type and concentration-dependent, the higher surface expression of TLR4 in $D_3$-MDMs found in our study could be potentially worrisome. Importantly, however, we have previously shown that TLR4 agonist LPS did not induce TNF-α production in $D_3$-MDMs [17], suggesting that signaling downstream TLR4 is affected by VitD3. Future studies should elucidate how VitD3 regulates the function of the TLR4 and TLR2 axis in macrophages in the presence and absence of DENV infection.

TLR9 recognizes CpG DNA, which is not produced by DENV replication, yet the expression of this TLR is upregulated in DCs infected with DENV-2 in vitro [28] and DCs obtained from DENV-infected patients [25]. We observed upregulation in TLR9 expression in MDMs induced by DENV-2 infection both at the mRNA and protein level. Intriguingly, this expression was found decreased in $D_3$-MDMs. This data is in line with other studies in which VitD3 was shown to decrease TLR9 expression in HeLa cells infected with Herpes simplex virus-1 [47], in monocytes [48], in murine plasmacytoid DCs [49], and in DENV-2 infected Mo-DCs [50]. We also found that VitD3 reduced the production of ROS at 2 hpi in $D_3$-MDMs after infection with DENV-2. Interestingly, when ROS production was inhibited with the ROS scavenger, NAC, in DENV-2-infected MDMs, a reduction in ROS production and TLR9 mRNA was observed, suggesting a direct upregulation of TLR9 expression by ROS production. To our knowledge, this is the first report that demonstrates VitD3 induced regulation of ROS production and therefore TLR9 downregulation in DENV-2 infected macrophages. Although similar results were observed in Mo-DCs, in which upregulation of TLR9 expression was driven by DENV-2 induction of ROS [28], TLR9 expression kinetics were different compared to our results. Further studies are needed to better understand this process and its relevance in the context of DENV infection.

Other studies have shown that VitD3 can protect from the effect of oxidative stress induced by ROS, especially H2O2, including apoptosis and inflammation in epithelial [51] and endothelial cells [52]. A novel mechanism by which VitD3 exerts its antioxidant functions involves activation of the Nrf2 pathway in mice [53] and diabetic rats [54]. Furthermore, VitD3 derivatives protect human keratinocytes from DNA damage and oxidative stress by upregulating Nrf2 and various antioxidant genes [55]. Whether VitD3 decreases the oxidative response of $D_3$-MDMs to DENV-2 infection by upregulating antioxidant genes driven by Nrf2, remains to be tested.

Following DENV-2 infection, SOCS-1 was upregulated in both $D_3$-MDMs and MDMs, but this increase was higher in $D_3$-MDMs. In neutrophils treated with VitD3 and infected with Streptococcus pneumoniae, there is an increased expression of SOCS-1 and SOCS-3 compared with non-treated neutrophils [56]. Further, mice treated with a synthetic analog of VitD3 upregulated the expression of SOCS-1 in a miR-155-dependent mechanism [21]. Accordingly, our results may imply that upregulation of SOCS-1 and downregulation of TLRs in DENV-2 infected $D_3$-MDMs could be contributing to the decreased inflammatory response, in addition to a decreased C-type lectin activation that was reported by us previously [17].

Finally, we did not observe any upregulation in IFN-I expression induced by VitD3 in DENV-2-infected $D_3$-MDMs. In contrast, we observed the downregulation of IFN α expression at 24 hpi, possibly due to a lower degree of replication of DENV-2. This suggests that the antiviral effect of VitD3 against DENV-2 in $D_3$-MDMs is independent of IFN I expression. Few studies have evaluated the regulation of antiviral responses induced by VitD3 in viral infections or other types of infectious diseases. Treatment of Huh7.5 hepatoma cells with the active form of VitD3 restricted HCV replication and induced upregulation of IFN β and the MxA antiviral gene [12]. Here, we observed a significant upregulation in PKR and OAS1

expression at 24 hpi in $D_3$-MDMs. Our results are in line with a study by Jadhav NJ et al, in which treatment with VitD3 of DENV-2 infected U937 DC-SIGN upregulated OAS1, OAS2, and OAS3 expression [15]. Altogether, previous reports and our results, indicate that VitD3 enhances the activity of IFN I rather than increasing its expression. VitD3 can promote the function of IFN I by fast non-genomic action. VDRs can interact directly with STAT1 and, therefore, somehow restrict the function of the JAK-STAT signaling pathway [57]. Indeed, VitD3 can compete with STAT1 for its natural receptor, VDR, and enhance the activity of IFN I by promoting the function of the JAK-STAT signaling pathway [29]. However, future studies are needed to explore the interaction between VDR and STAT1 in DENV-2-infected MDMs and its implications in VitD3 enhancement of IFN I activity.

Although the present study is an attempt at decoding the role of vitamin D in dengue disease pathogenesis, one more limitation worth noting is that the results were obtained only with DENV-2. Although Puerta-Guardo et al. [16] showed that VitD3 decreased both infections of DENV-4 in U937 and Huh-7 cells and levels of proinflammatory cytokines, no studies have addressed yet the effect of VitD3 in DENV-1 and -3 replication. However, as most of the effects of VitD3 occurred at the cellular level (in vitro), the same results can likely be observed during infection with other DENV serotypes.

In conclusion, our study demonstrates the immunoregulatory effects of VitD3 in DENV-2-infected $D_3$-MDMs. The VitD3-mediated modulation of components of the innate immune response, such as expression of TLRs, production of ROS, and expression of SOCS-1 during DENV-2 infection, thus contributing to the anti-inflammatory effects of this hormone (Fig 7).

## Supporting information

**S1 Fig. Purity of monocyte and monocyte-derived macrophages (MDMs) cultures.** Monocytes were purified from PBMCs, and purity was assessed by the quantification of contaminant CD19+, CD3+ and CD56+ (A, B and C). In D, a representative histogram with CD68 expression levels of monocyte-derived macrophages (MDMs) and $D_3$-MDMs after 6 days of differentiation is shown. Figure in C represent 3 individual experiments.
(TIF)

**S2 Fig. VitD3 reduces DENV-2 infection and replication in D3-MDMs.** MDMs were differentiated in the presence of VitD3 (0.1 nM) for 6 days (D3-MDMs) and infected with DENV-2 with an MOI of 5 for 24 h. Infection was evaluated through flow cytometry and expressed as % E-positive cells. In A, a representative dot-plot of an individual experiment is shown, while B shows a summary of the results. Replication of DENV-2 was evaluated by quantification of intracellular viral RNA copy number using qPCR (C) and by quantification of viral titer in the supernatant using a plaque assay (D). Figures represent 10 individual experiments from different donors (B), while five individual experiments are shown in C. Differences were obtained with a Wilcoxon test using a 95% confidence interval ($^{**}p < 0.01$, $^{*}p < 0.05$).
(TIF)

**S3 Fig. VitD3 reduces the production of IL-6, TNF-α, and IL-10 in $D_3$-MDMs.** MDMs were differentiated in the presence of VitD3 (0.1 nM) for 6 days ($D_3$-MDMs) and then infected with DENV-2 with a MOI of 5 for 2, 8, or 24h. Production of IL-6 (A), TNF-α (B), and IL-10 (C) was measured by ELISA in MDMs and $D_3$-MDMs. Figures represent six individual experiments from different donors (A and B), while four individual experiments are shown in C. Differences were obtained using a two-way ANOVA with a Bonferroni post-hoc test using a 95% confidence interval ($^{***}$ p < 0 .001)
(TIF)

**S4 Fig. Expression of TLRs, IFN-I, ISGs and SOCS-1 at 2 8 and 24 hours in mock D3-MDMs.** MDMs were differentiated in the presence of VitD3 (0.1 nM) for 6 days (D3-MDMs) and then expression of RIG-I, TLR7, TLR3, IFN-α, IFN-β, PKR and OAS1 were measured by qPCR in MDMs and D3-MDMs using ubiquitin-conjugating enzyme E2D2 as housekeeping. Data is expressed as fold change relative to mock treated MDMs. Figures represent three individual experiments from different donors. Differences were obtained with a two-way ANOVA with a Bonferroni post-hoc test and a 95% confidence interval (***p < 0.001, **p < 0.01, *p < 0.05).
(TIF)

**S5 Fig. VitD3 regulates the expression of CYP24A1, VDR, TLR3 and TLR4 in D3-MDMs, but not TLR8.** MDMs were differentiated in the presence of VitD3 (0.1 nM) for 6 days (D3-MDMs) and then infected with DENV-2 with a MOI of 5 for 24 h. Expression of CYP24A1 (A) and TLR8 (D) were measured by qPCR in MDMs and D3-MDMs both infected and mock-infected with DENV-2, using the gene encoding ubiquitin-conjugating enzyme E2D2 as a housekeeper gene. Data is expressed as relative expression normalized to the expression level of the housekeeper gene. Expression at the protein level of VDR (B), TLR3 (E) TLR4 (F) and TLR9 (G) was measured by flow cytometry and expressed as mean fluorescence intensity. Figures represent five individual experiments from different donors. Differences were obtained with a two-way ANOVA with a Bonferroni post-hoc test and a 95% confidence interval (**p < 0.01, *p < 0.05).
(TIF)

**S1 Table. Primers used in the study**
(DOCX)

## Acknowledgments

The authors would like to thank the blood bank of the "Escuela de Microbiologia, UdeA, Medellín Colombia" for providing us with leukocyte-enriched blood units from healthy individuals and the personnel at the institutions where the study was performed. Fig 7 was created with BioRender.com.

## Author Contributions

**Conceptualization:** Jorge Andrés Castillo, Jolanda M. Smit, Izabela A. Rodenhuis-Zybert, Silvio Urcuqui-Inchima.

**Data curation:** Jorge Andrés Castillo, Jolanda M. Smit, Izabela A. Rodenhuis-Zybert, Silvio Urcuqui-Inchima.

**Formal analysis:** Jorge Andrés Castillo, Diana M. Giraldo, Jolanda M. Smit, Izabela A. Rodenhuis-Zybert, Silvio Urcuqui-Inchima.

**Funding acquisition:** Silvio Urcuqui-Inchima.

**Investigation:** Jorge Andrés Castillo, Diana M. Giraldo.

**Methodology:** Jorge Andrés Castillo, Diana M. Giraldo, Silvio Urcuqui-Inchima.

**Project administration:** Silvio Urcuqui-Inchima.

**Supervision:** Silvio Urcuqui-Inchima.

**Writing – original draft:** Jorge Andrés Castillo, Diana M. Giraldo, Silvio Urcuqui-Inchima.

**Writing – review & editing:** Jorge Andrés Castillo, Diana M. Giraldo, Juan C. Hernandez, Jolanda M. Smit, Izabela A. Rodenhuis-Zybert, Silvio Urcuqui-Inchima.

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
