## [Decision Letter · Decision Letter 0]

26 Aug 2021

Dear Professor Urcuqui-Inchima,

Thank you very much for submitting your manuscript "Regulation of innate immune responses in macrophages differentiated in the presence of vitamin D and infected with dengue virus 2" for consideration at PLOS Neglected Tropical Diseases. As with all papers reviewed by the journal, your manuscript was reviewed by members of the editorial board and by several independent reviewers. The reviewers appreciated the attention to an important topic. Based on the reviews, we are likely to accept this manuscript for publication, providing that you modify the manuscript according to the review recommendations. 

The manuscript was evaluated by two specialists in the area which had a positive impression about the manuscript, but still suggested minor changes. Please, respond the suggestions and queries of the reviewers as fully as possible. We ask especially to address the points of further clarification.

Sincerely,

Helton da Costa Santiago, M.D., Ph.D

Associate Editor

Rebecca Rico-Hesse

Deputy Editor

The manuscript was evaluated by two specialists in the area which had a positive impression about the manuscript, but still suggested minor changes. Please, respond the suggestions and queries of the reviewers as fully as possible. We ask especially to address the points of further clarification.

Reviewer's Responses to Questions

**Key Review Criteria Required for Acceptance?**

**Methods**

-Are the objectives of the study clearly articulated with a clear testable hypothesis stated?

-Is the study design appropriate to address the stated objectives?

-Is the population clearly described and appropriate for the hypothesis being tested?

-Is the sample size sufficient to ensure adequate power to address the hypothesis being tested?

-Were correct statistical analysis used to support conclusions?

-Are there concerns about ethical or regulatory requirements being met?

Reviewer #1: yes

Reviewer #2: -Are the objectives of the study clearly articulated with a clear testable hypothesis stated? 

Yes

-Is the study design appropriate to address the stated objectives? 

Yes

-Is the population clearly described and appropriate for the hypothesis being tested? 

Yes

-Is the sample size sufficient to ensure adequate power to address the hypothesis being tested? 

Yes

-Were correct statistical analysis used to support conclusions? 

No. Statistical analysis needs to be revised.

-Are there concerns about ethical or regulatory requirements being met?

Yes

**Results**

-Does the analysis presented match the analysis plan?

-Are the results clearly and completely presented?

-Are the figures (Tables, Images) of sufficient quality for clarity?

Reviewer #1: Ok, see attached comments.

Reviewer #2: -Does the analysis presented match the analysis plan?

Yes

-Are the results clearly and completely presented?

No

-Are the figures (Tables, Images) of sufficient quality for clarity?

Yes

**Conclusions**

-Are the conclusions supported by the data presented?

-Are the limitations of analysis clearly described?

-Do the authors discuss how these data can be helpful to advance our understanding of the topic under study?

-Is public health relevance addressed?

Reviewer #1: ok, see attached comments

Reviewer #2: -Are the conclusions supported by the data presented?

Yes

-Are the limitations of analysis clearly described?

No

-Do the authors discuss how these data can be helpful to advance our understanding of the topic under study?

Yes

-Is public health relevance addressed?

Not clearly

**Editorial and Data Presentation Modifications?**

Reviewer #1: ok

Reviewer #2: 1. METHODS

a) Statistical analyses are poorly described and need to be revised. There is no mention of the Mann-Whitney test used in Fig. S1 and Fig. 2 and the normality assessment that justify the employment of such unpaired non-parametric test. In Fig. S1, the use of the Mann-Whitney test for matched-pair comparisons is inappropriate. Also, authors should check the use of two-way ANOVA to compare mean differences between MDMs vs. D3-MDMs at the same time point. 

b) Flow cytometry was used to assess the expression pattern of VDR and DHR beyond TLR3/4/9 and DENV E antigen. This needs to be mentioned.

c) In table S1, use the gene nomenclature to name PCR targets for which primers were designed.

2. RESULTS

a) The authors have done a ton of fine work, but in my opinion, they should improve the way the results are presented in order to make them easier to understand. My suggestions are: (i) standardize supplementary and main figures, plotting mean and SEM (or SD) plus individual values; (ii) label title to figures and then make visual separation of qPCR and flow cytometry results clearly. Labeling Y-axis differently [e.g. PKR/Ubiquitin mRNA (Fold-change of mock controls)] may be useful; (iii) present dot- and/or MFI-plot along with flow cytometry results; (iv) consider including mock-infected controls in each graph as done in panel 1D; (v) clarify in figure legends what each statistical hypothesis was being tested for; and (vi) indicate statistically and non-statistically significant differences in each figure panel (including S1D). 

b) Empirical data demonstrating the purity of MDMs is missing. Authors should include it as supplementary material.

c) Why did the authors use only MFI to infer protein levels? Is there any difference concerning the number of positive cells for a given target between MDMS and D3-MDMs? 

d) 3rd paragraph “ …observed between these types of cells”: they are all macrophages.

e) Fig 1A and 1D: How do the authors explain the early reduction of TLR3 after 2hpi with DENV-2 since mRNA level is changed by 24hpi? 

f) Mention is made of ELISA assays for IL-6, IL-10, and TNF but no data are presented in the manuscript. I would really like to see this data. It would confirm the results previously published by the group (Ref. 17) and reinforce the main claims of this study.

3. DISCUSSION

a) Could the data collected for DENV-2 be extended for other DENV serotypes? Authors should comment on this. 

b) There is no evidence that the authors see any limitation to their own work. They should add a paragraph on limitations of the work presented.

**Summary and General Comments**

Reviewer #1: In the present work, Castillo JA., et al. investigated the effect of Vit-D3 on the innate immune responses against DENV-2. They found that replication of DENV-2 in VitD3 treated monocyte-derived macrophages was restricted. They showed that VitD3 treatment downregulated the expression levels of TLRs, RIG but increased levels of SOCS-1, reduced ROS production related to the lower expression of TLR-9. Surprisingly, VitD3 did not modulate expression of IFN-α and IFB-β but higher expression of PKR and OAS1 mRNA were found in VitD3-MDM. 

Comments

1. Fig. 1 authors monitored the expression levels of RIG-I, TLR-3 and TLR-7 mRNA and the levels of TLR3, TLR7 protein productions. As shown in Fig.1C, mRNA of TLR-3 were peaked at 8 hr of infection in MDM treated and untreated with VitD3 while there was no change of TLR3 at 2 h of infection. Surprisingly, TLR-3 protein production peaked at 2 hr of infection (Fig. 1D). How can protein production reach the highest level earlier than its mRNA expression? This point need to be clarified.

2. Fig.2 means of % suppression at each MOI should be added in the Result section. In addition, differences of those means should be statistically compared among MOI. 

3. On page 8, 2nd paragraph, Fig. 3A-B, authors conclude that VitD3 regulated expression of TLRs independently of the MOI used. ……………..Taken together, our results suggest that…………………….in response to DENV-2 infection, possibly depending on the time of infection (early for TLR3, and later for TLR7). This point should be clarified. The author concluded that the response does not dependent on concentration of infection dose but depends on time of infection. The explanation should be added such as potential mechanism.

4. In Discussion, a potential diagram showing how VitD3 suppress innate responses should be included.

Reviewer #2: This study examines the immunomodulatory effects of bioactive vitamin D3 metabolite (Vit.D3) in DENV-2-infected human macrophages. The authors show that monocytes differentiated into macrophages (MDMs) in the presence of VitD3 exhibit decreased expression of pattern recognition receptors (e.g. TLR3, TLR7, RIG-I) and elevated transcriptional activity of some IFN-stimulated genes, upon DENV-2 challenge. Overall, this is a strong study that reinforces the immunomodulatory role of vitamin D. However, there are some aspects of it that appear to show a lack of rigor including the statistical analysis. Once these are dealt with appropriately the manuscript would be far more convincing.

PLOS authors have the option to publish the peer review history of their article (what does this mean?). If published, this will include your full peer review and any attached files.

Reviewer #1: No

Reviewer #2: No

Figure Files:

Data Requirements:

Reproducibility:

References

---

## [Decision Letter · Decision Letter 1]

5 Oct 2021

Dear Professor Urcuqui-Inchima,

We are pleased to inform you that your manuscript 'Regulation of innate immune responses in macrophages differentiated in the presence of vitamin D and infected with dengue virus 2' has been provisionally accepted for publication in PLOS Neglected Tropical Diseases.

Best regards,

Helton da Costa Santiago, M.D., Ph.D

Associate Editor

Rebecca Rico-Hesse

Deputy Editor

Reviewer's Responses to Questions

**Key Review Criteria Required for Acceptance?**

**Methods**

-Are the objectives of the study clearly articulated with a clear testable hypothesis stated?

-Is the study design appropriate to address the stated objectives?

-Is the population clearly described and appropriate for the hypothesis being tested?

-Is the sample size sufficient to ensure adequate power to address the hypothesis being tested?

-Were correct statistical analysis used to support conclusions?

-Are there concerns about ethical or regulatory requirements being met?

Reviewer #2: Yes

**Results**

-Does the analysis presented match the analysis plan?

-Are the results clearly and completely presented?

-Are the figures (Tables, Images) of sufficient quality for clarity?

Reviewer #2: Yes

**Conclusions**

-Are the conclusions supported by the data presented?

-Are the limitations of analysis clearly described?

-Do the authors discuss how these data can be helpful to advance our understanding of the topic under study?

-Is public health relevance addressed?

Reviewer #2: Yes

**Editorial and Data Presentation Modifications?**

Reviewer #2: (No Response)

**Summary and General Comments**

Reviewer #2: (No Response)

PLOS authors have the option to publish the peer review history of their article (what does this mean?). If published, this will include your full peer review and any attached files.

Reviewer #2: No

---

## [Editor Report · Acceptance letter]

7 Oct 2021

Dear Professor Urcuqui-Inchima,

We are delighted to inform you that your manuscript, "Regulation of innate immune responses in macrophages differentiated in the presence of vitamin D and infected with dengue virus 2," has been formally accepted for publication in PLOS Neglected Tropical Diseases.

Best regards,

Shaden Kamhawi

co-Editor-in-Chief

Paul Brindley

co-Editor-in-Chief
